# Pre-Exposure of Early-Weaned Lambs to a Herb-Clover Mix Does Not Improve Their Subsequent Growth

**DOI:** 10.3390/ani10081354

**Published:** 2020-08-05

**Authors:** Lukshman Jay. Ekanayake, Rene Anne Corner-Thomas, Lydia Margaret Cranston, Paul Richard Kenyon, Stephen Todd Morris, Sarah Jean Pain

**Affiliations:** 1Department of Animal Science, Faculty of Agriculture, University of Peradeniya, Peradeniya 20000, Sri Lanka; 2School of Agriculture and Environment, Private Bag 11-222, Massey University, Palmerston North 4442, New Zealand; R.Corner@massey.ac.nz (R.A.C.-T.); L.Cranston@massey.ac.nz (L.M.C.); P.R.Kenyon@massey.ac.nz (P.R.K.); S.T.Morris@massey.ac.nz (S.T.M.); S.J.Pain@massey.ac.nz (S.J.P.)

**Keywords:** early weaning, lamb, rumen development, herb–clover mix

## Abstract

**Simple Summary:**

Exposure of lambs to herbage-based diets prior to weaning may facilitate the development of the rumen which may subsequently increase animal performance after early weaning. The aim of this study was to estimate the effects of varying durations of exposure of lambs to a herb–clover mix containing chicory, plantain, red clover, and white clover prior to early weaning (at ~45 days of age) on their subsequent growth and rumen development at conventional weaning age. Prolonged exposure of lambs to the herb–clover mix prior to early weaning had no impact on lamb growth or rumen development, suggesting that using this management option will not improve performance of lambs after early weaning.

**Abstract:**

Twin sets of lambs were randomly allocated to one of six treatments: (1) lambs born and managed on ryegrass–clover-based pasture until conventional weaning approximately at 99 days of age (Grass–Grass_CW_); (2) lambs born on ryegrass–clover-based pasture and early weaned onto a herb–clover mix at ~45 days of age (Grass–Herb_EW_); (3) lambs born on ryegrass–clover-based pasture, transferred with their dam onto a herb–clover mix at ~45 days of age until conventional weaning (Grass–Herb_CW_); (4) lambs born on ryegrass–clover-based pasture, transferred with their dam onto a herb–clover mix at ~15 days of age and early weaned onto a herb–clover mix at ~45 days of age (Grass–Herb_D15EW_); (5) lambs born and managed on herb–clover mix until conventional weaning (Herb–Herb_CW_); (6) lambs born on herb–clover mix and weaned early onto a herb–clover mix at ~45 days of age (Herb–Herb_EW_). In both years, Herb–Herb_CW_ lambs had greater (*p* < 0.05) growth rates than lambs in other treatments. The liveweight gains and rumen papillae development of Herb–Herb_EW_, Grass–Herb_D15EW_ and Grass–Herb_EW_ lambs did not differ (*p* > 0.05). The weight of the empty digestive tract components at either early weaning or conventional weaning did not differ (*p* > 0.05) between treatments. Exposing early-weaned lambs to the herb mix for a prolonged period, prior to early weaning, does not improve their subsequent growth.

## 1. Introduction

Weaning lambs early can be a useful management option that farmers can use when herbage production is inadequate or of poor quality, to reduce the overall ewe feed demand and to support the growth of early-weaned lambs to achieve liveweight targets [1,2]. Lambs weaned early at 4–8 weeks of age onto a ryegrass–clover-based pasture have consistently displayed lower live weight gains and poorer survival compared to unweaned lambs on ryegrass–clover-based pastures, to approximately 14 weeks of age [3,4]. This is likely explained by the inadequate nutritional quality of ryegrass–clover-based pastures. In contrast, a herb–clover mix containing plantain (*Plantago lanceolata*), chicory (*Cichorium intybus*), red clover (*Trifolium pretanse*) and white clover (*Trifolium repens*) has been shown to have a greater nutritional quality compared to a ryegrass–clover-based pasture [5,6,7]. This suggests that a herb–clove mix can be used to support lamb growth, post-early weaning. However, the liveweight gain of lambs weaned early, at a minimum live weight of 14 to 16 kg, onto a herb–clover mix, has been inconsistent across studies, compared to unweaned lambs on ryegrass–clover-based pastures [8,9,10,11]. In contrast, the live weight of ewes whose lambs were weaned early has consistently been greater than those left unweaned on ryegrass–clover [10,11]. The inconsistency in lamb liveweight gain has not yet been explained. The reasons for the variation in lamb growth post-early weaning on a herb–clover mix appear not to be driven by variation of metabolisable energy [9,11] or crude protein content [9,10,11,12] of the herbage. If farmers are to utilize early weaning, a consistent method for achieving suitable lamb growth post weaning is required. The impact of prior exposure to the herb–clover mix on lamb growth, post early weaning, has not been previously examined. If prior exposure to the herb–clover mix was found to result in a consistent positive effect on lamb growth, management guidelines for farmers could be developed.

Early development of the reticulo-rumen and its microbial population is important for efficient utilization of herbage-based diets after weaning [13,14]. Early exposure to herbage-based diets can help develop a strong host rumen microbiota [15,16] and familiarity with a herbage type can influence a lamb’s preference for that herbage later in life [17]. Sheep avoid grazing novel herbages [18,19] for several reasons including taste aversion [20], as a strategy to maintain effective rumen micro-flora [21] and to meet their ideal protein intake [22,23]. Combined, these studies suggest that early exposure of lambs to herbage may facilitate rumen development and increase subsequent animal performance. To date the rumen development of early-weaned lambs offered herb–clover mix has not been studied. It was hypothesized that prolonged exposure of lambs to a herb–clover mix, prior to early weaning, would have a positive effect on their subsequent growth and rumen development. The aims of the present study were to test the effects of (i) weaning lambs early at approximately 45 days of age onto a herb–clover mix and (ii) early prolonged exposure of lambs to a herb–clover mix prior to early weaning at approximately 45 days of age on their subsequent growth and rumen development until conventional weaning age (~99 days of age).

## 2. Materials and Methods

### 2.1. Herbage Treatments

Eight paddocks (12.85 ha in total land area) of herb–clover mix and seven paddocks (14.85 ha in total land area) of ryegrass–clover-based pasture containing perennial ryegrass (*Lolium perenne* L.) and white clover (*Trifolium repens*) were used over two spring periods in 2017 and 2018. Herb–clover mix paddocks were sown during autumn in 2012 and 2013 with a seed mixture of chicory (*Cichorium intybus*; 6 kg/ha), plantain (*Plantago lanceolata*; 6 kg/ha), white clover (4 kg/ha) and red clover (*Trifolium pratense*; 6 kg/ha). During the experimental period (August to December in both years), a rotational grazing system was used whereby the post-grazing sward surface heights were maintained at a minimum of 5 to 7 cm for both ryegrass–clover-based pasture and herb–clover mix, respectively, in order to provide ad-libitum dry matter intakes for lambs and ewes [24,25]. The management of the herbages in this study was similar to the management applied in previous studies [8,9,10,11,12].

### 2.2. Experimental Design

The studies were conducted at Massey University’s Keeble farm, 5 km southeast of Palmerston North, New Zealand (40°24′ S and 175°36′ E). All animal manipulations were approved by the Massey University Animal Ethics Committee (MUAEC 17-40). Each year, Romney ewes, from a large commercial flock, were naturally bred during a 17-day breeding period. A random subset of ewes diagnosed as twin-bearing using transabdominal ultrasound at approximately 90 days of pregnancy, were included in each year. Throughout the gestation period, within each year, ewes were managed as a single flock (group) under commercial pastoral farming conditions on a ryegrass–clover-based pasture.

Each year, twin-bearing ewes were allocated to treatments three days prior to lambing using a stratified random sampling procedure, in order to balance the groups for ewe live weight. A summary of the experimental design can be found in Table 1. Briefly, in 2017, the study included six treatments (Grass–Grass_CW_, Grass–Herb_EW_, Grass–Herb_CW_, Grass–Herb_D15EW_, Herb–Herb_CW_ and Herb–Herb_EW_, Table 1), while in 2018, Grass–Grass_CW_ was omitted resulting in five treatments (Grass–Herb_EW_, Grass–Herb_CW_, Grass–Herb_D15EW_, Herb–Herb_CW_ and Herb–Herb_EW_). The Grass–Grass_CW_ treatment had previously been examined over three years [10,11] and was determined to be unnecessary as the overall aim was to evaluate the effects of early exposure of lambs to a herb–clover mix prior to early weaning on their subsequent growth and rumen development until a conventional weaning age. Three days prior to lambing, ewes (*n* = 117 in 2017 and *n* = 161 in 2018) were allocated to Grass–Herb_EW_, Grass–Herb_CW_, and Grass–Herb_D15EW_ and Grass–Grass_CW_ (only in 2017) treatments and grazed on a ryegrass–clover-based pasture. Similarly, on the same date, ewes (*n* = 80 in 2017 and *n* = 82 in 2018) were allocated to Herb–Herb_CW_ and Herb–Herb_EW_ treatments begun grazing the herb–clover mix.

Lambing began on 28 and 27 August in 2017 and 2018 (L0), respectively. Lambs were born over 26- and 28-day period in 2017 and 2018 (mid-point of the lambing period each year was defined as L1, i.e., average day of age at that point was one). All lambs were weighed, ear tagged and identified to their dam within 24 h of birth. In total, *n* = 136 and *n* = 138 lambs were born on herb–clover mix in 2017 and 2018, respectively. On the ryegrass–clover-based pasture *n* = 224 and *n* = 302 lambs were born in 2017 and 2018, respectively. In accordance with veterinary advice, at L27 (average 27 days of age) in 2017 and L26 in 2018, lambs were drenched with an oral triple combination drench (Matrix, Merial Ancare, Manukau City, New Zealand) and thereafter at 28-day intervals throughout the study at a rate of 1 mL per 5 kg live weight to control internal parasites. This method has been successfully used in previous studies [8,9,10,11], therefore, lamb growth in this study was unlikely to be affected by internal parasites.

At L15 (average 15 dayss of age) in both years, ewes and lambs allocated to the Grass–Herb_D15EW_ treatment were transferred from ryegrass–clover-based pasture to a herb–clover mix where they remained until early weaning (at L51 and L44 in 2017 and 2018, respectively, Table 1). Ewes and lambs allocated to the remaining treatments were maintained on their respective herbage until early weaning. At L51 in 2017 and at L44 in 2018 (early weaning), ewes rearing twin lambs (*n* = 123 and *n* = 112 in 2017 and 2018, respectively) that were both a minimum of 14 kg live weight (*n* = 246 and *n* = 224, in 2017 and 2018, respectively) were included in the study. Any ewes and lambs that did not fulfil these criteria were excluded from the remainder of the study. Lambs in each treatment (*n* = 42, 44, 38, 42, 46, 34 in Herb–Herb_CW_, Herb–Herb_EW_, Grass–Herb_D15EW_, Grass–Herb_CW_, Grass–Herb_EW_, Grass–Grass_CW_ in 2017 and *n* = 36, 36, 42, 34, 34 in Herb–Herb_CW_, Herb–Herb_EW_, Grass–Herb_D15EW_, Grass–Herb_CW_, Grass–Herb_EW_ in 2018, respectively) remained on these treatments until conventional weaning at L99 in 2017 and L87 in 2018.

After early-weaning, ewes in the Grass–Herb_EW_, Grass–Herb_D15EW_ and Herb–Herb_EW_ treatments (weaned ewes) were managed with Grass–Grass_CW_ treatment in 2017 until L99 or with a commercial flock under pastoral farming conditions on ryegrass–clover-based pasture until L87 in 2018, receiving similar allowances in terms of pasture offered. Early-weaned lambs (Grass–Herb_EW_, Grass–Herb_D15EW_ and Herb–Herb_EW_) were managed with the Herb–Herb_CW_ ewes and lambs until L99 and L87 in 2017 and 2018, respectively. Within 1 to 2 h of birth, ewes develop an exclusive bond with their lambs and reject any alien lamb that attempts to suck for the remainder of the lactation [26]. Therefore, there was no concern regarding the early-weaned lambs suckling from unweaned ewes.

### 2.3. Animal Measurements

Lambs and ewes were weighed within 1 h of removal from herbage at L15, L51, L65 and L99 in 2017 and L15, L44, L61 and L87 in 2018. Ewe body condition score (BCS, scale 1–5, including half units, [27]) was assessed at each weighing by a single experienced operator.

### 2.4. Anatomical and Histological Examination of the Digestive System

A subset of male lambs from the complete twin sets were selected for anatomical and histological examination of the digestive system at L44 (*n* = 12) and L87 (*n* = 24) in 2018 to determine the effect of feed type prior to early weaning, and the combination of feed type and early weaning on the rumen development, respectively. Lambs were selected using stratified random sampling procedure, based on live weights, in order to obtain a representative sample of lambs from each treatment. At L44, lambs in the Grass–Herb_EW_ (*n* = 6, average live weight 19.0 ± 1.5) and Herb–Herb_EW_ (*n* = 6, average live weight 21.0 ± 1.5) treatments were euthanised. At L87, 24 male lambs were selected from the Grass–Herb_EW_ (*n* = 6; average live weight 27.4 ± 2.0), Grass–Herb_CW_ (*n* = 6; average live weight 31.0 ± 2.0), Herb–Herb_CW_ (*n* = 6; average live weight 35.0 ± 2.0) and Herb–Herb_EW_ (*n* = 6; average live weight 32.0 ± 2.0) treatments and euthanised. Lambs were weighed within 1 h of removal from herbage and were euthanised using captive bolt stunning and exsanguination.

Immediately after euthanasia, their skins removed, they were eviscerated, and the head was separated from the carcass. Hot carcass weight was then recorded. The digestive system was removed by severing the esophagus which was tied at the cardia of the stomach. All the organs attached to the digestive system were removed and then weighed. The digestive system was then separated into its components (reticulo-rumen, small intestine and large intestine) which were cleaned using tap water. Empty component weights were recorded to the nearest gram. The cleaned reticulo-rumen was dissected according to the procedure of [28] and the reticulo-rumen was opened and laid flat symmetrically. Two samples (1 cm^2^ each) were collected from the right side of the caudal dorsal sac as [29] reported that samples from this region were representative of rumen development.

Rumen tissue samples (two from each animal) were immediately fixed in a 10% formalin saline solution for 24 h. Samples were then processed using a Excelsior ES Tissue Processor (ThermoFisher^®^, Waltham, MA, USA). Dehydration was carried out by ascending graded series of ethyl alcohol (70, 95% and absolute Alcohol) for 12 h, followed by clearing in Xylene at 60 °C. The samples were then embedded and impregnated in melted Paraffin wax at 55–58 °C using a HistoStar Embedder (ThermoFisher^®^,MA, USA). Embedded samples were cut using Rotary Microtome (MicroTec^®^, Wetzlar, Germany) into sections (4 µm). Two sections were produced, 150 µm apart from each tissue sample resulting in 4 sections per animal (replicates). The sections were floated in a Tissue Bath (ThermoFisher^®^, MA, USA) at 43 °C and were mounted onto adhesive pre-cleaned slides (90° ground edges, 76 mm × 26 mm). The sections were stained using Haemytoxelin and Eosin on an Autostainer XL (Leica^®^, Wetzlar, Germany). The stained slides, 2 slides per lamb, each contained two sections were covered using a Leica^®^ CV5030 cover slip. All complete papillae in each section were examined microscopically to determine their length and width at the base (Figure 1).

### 2.5. Herbage Measurements

Herbage masses were measured at L15, L51 and L99 in 2017 and L15, L44 and L87 in 2018. Four random quadrat cuts (0.1 m^2^ each) were taken to ground level from each herbage type at each sampling date using an electric shearing handpiece [30]. Samples were then oven dried to a constant weight to estimate herbage mass. In addition, four composite herbage samples, each containing ten grab samples per herbage type, were also collected, to mimic post grazing height, at each sampling date to determine the botanical and nutritional composition [30]. To determine the botanical composition, a subsample from each composite sample (4 per herbage type) was sorted into each species (herb–clover mix: plantain, chicory, red clover, white clover; ryegrass–clover-based pasture: ryegrass, clover; other grasses (combined), weeds, and dead matter) and then oven dried and weighed to determine the botanical composition. The remaining sample was then frozen, dried, ground, sieved (1 mm) and analysed using in vitro methods to determine the nutritional quality. These measures included dry matter digestibility (DMD, [31]), digestible organic matter digestibility (DOMD, [31]), percentage crude protein (CP; “Dumas” procedure, AOAC method 968.06 using a Leco total combustion method, LECO Corporation, St. Joseph, MI, USA). Percentage acid detergent fibre (ADF) was analysed by a Tecator Fibretec System [32]. Metabolisable energy (ME) content of herbages was calculated from the organic matter digestibility (DOMD × 0.16 MJ/Kg DM, [31]). These herbage measurements have been used in previous studies [8,9,10,11,12] to estimate the dry matter mass, botanical and chemical composition of both ryegrass–clover-based pasture and herb–clover mix.

### 2.6. Statistical Analysis

The individual animal was considered the experimental unit for these analyses. Live weight of lambs and ewes were subjected to analysis of variance for repeated measures using the MIXED procedure in SAS (Statistical Analysis System, version 9.2; SAS Institute Inc., Cary, NC, USA). Analyses were performed separately for each year due to the differences in the days on which measurements were collected, number of treatments in 2017 and 2018, differences in climate and herbage quality, and different animals used between years. The model for lamb live weight at L51 and L99 and at L44 and L87 in 2017 and 2018, respectively, included the fixed effects of weaning treatment, sex of lamb (male, female), measurement date and the two-way interactions of treatment and measurement date. The live weight of lambs at the start of the treatment period was included in the model as a covariate. The model for lamb liveweight gain from early weaning to conventional weaning included the fixed effect of weaning treatment and sex of lamb.

The models for ewe live weight included the fixed effects of weaning treatment, measurement date and two-way interactions of treatment and measurement date. The model for ewe liveweight gain from early weaning to conventional weaning included the fixed effect of weaning treatment. The live weight of the ewe at the start of the treatment period was included in the model as a covariate. Ewe body condition score was analysed using a Poisson distribution and logit transformation using the GENMOD procedure in SAS. The Poisson distribution was chosen as it is a nonlinear regression model for discrete outcomes. The model included the fixed effects of weaning treatment, measurement date and the two-way interaction of treatment and measurement date.

Papillae length and width were subjected to analysis of variance using the MIXED procedure in SAS. The models for papillae length and width included the fixed effect of weaning treatment and the random effect of lamb, sample and replicate (sections) for each sample. The live weight of the lamb at slaughter was included in the model as a covariate. The models for the anatomical measurements of the rumen included the fixed effect of weaning treatment and lamb live weight at slaughter as a covariate.

The botanical composition of herbages was subjected to an analysis of variance for repeated measures using the MIXED procedure in SAS. The model included fixed effects of plant species and measurement date. Herbage masses were analysed using a model that included herbage type and measurement date as fixed effects. The nutritional quality data were analysed using the MIXED procedure in a model that included the fixed effects of herbage type and measurement date.

## 3. Results

### 3.1. Botanical Composition, Herbage Mass and Nutritional Quality of Herbage

The percentage of chicory and clover was 16% greater (*p* < 0.05) and plantain was 23% lower (*p* < 0.05) in the herb–clover mix at L44 in 2018 than at L47 in 2017 (Figure 2). The percentage of plantain was 20% greater (*p* < 0.05) and dead matter content was 35% lower (*p* < 0.05) in the herb–clover mix at L87 in 2018 than at L99 in 2017. Other plant components in the herb–clover mix did not differ (*p* > 0.05) between years and time points. At L15, the percentage of ryegrass in the ryegrass–clover-based pasture was 22% greater (*p* < 0.05) in 2018 than 2017. Other plant components in the ryegrass–clover-based pasture did not differ (*p* > 0.05) between years and time points. In both years, herbage mass of both herbage types was above 1600 kg DM/ha at all sampling dates throughout the study (Table 2).

At L15 in 2017, the CP content of ryegrass–clover-based pasture was ~5% greater (*p* < 0.05) than herb–clover mix whereas the reverse was seen at L51. At L99, CP content did not differ (*p* > 0.05) between herbages. At L15 and L99, the ADF content of herb–clover mix was greater (13% and 7%, respectively, *p* < 0.05) than that of ryegrass–clover-based pasture, however, at L51 ADF did not differ (*p* > 0.05). The DMD, neutral detergent fibre (NDF) and ME of herb–clover mix were lower (~6 to 10%, 14 to 22% and 1.1 to 1.8 MJ/Kg, respectively, *p* < 0.05) than ryegrass–clover-based pasture at all sampling times.

In 2018, at L15 and L87, the CP content of herb–clover mix was lower (~3 and 6%, respectively, *p* < 0.05) than ryegrass–clover-based pasture, whereas the CP content of herb–clover mix was ~3% greater (*p* < 0.05) at L44 (Table 2). The ADF content of herb–clover mix did not differ (*p* > 0.05) from that of ryegrass–clover-based pasture at L15 and L87 but was ~5% lower (*p* < 0.05) at L44. The NDF of herb–clover mix was 10 to 20% lower (*p* < 0.05) than that of ryegrass–clover-based pasture at L15, L44 and L87. The DMD and ME of herb–clover mix were greater (~4 to 8% and 0.4 to 1.2 MJ/Kg, respectively, *p* < 0.05) than that of ryegrass–clover-based pasture at L15, L44 and L87.

### 3.2. Lamb Live Weight and Liveweight Gain

In 2017, at L51 (date of early weaning), the live weights of lambs did not differ (*p* > 0.05) among treatments (Table 3). At L65, the live weight of lambs in the Herb–Herb_EW_, Grass–Herb_D15EW_ and Grass–Herb_EW_ treatments did not differ (*p* > 0.05) but were ~1 kg lighter (*p* < 0.05) than Herb–Herb_CW_, Grass–Herb_CW_ and Grass–Grass_CW_ lambs. At L65, Herb–Herb_CW_ lambs were 0.8 kg heavier (*p* < 0.05) than Grass–Grass_CW_ lambs but neither differed (*p* > 0.05) from Grass–Herb_CW_ lambs. At L99 (at conventional weaning), Herb–Herb_CW_ and Grass–Herb_CW_ lambs did not differ (*p* > 0.05) in live weight but were 4 to 5 kg heavier (*p* < 0.05) than lambs in all other treatments, which did not differ (*p* > 0.05) from one another. Lamb liveweight gains between L51 and L99 in the Herb–Herb_CW_ and Grass–Herb_CW_ treatments were greater (*p* < 0.05) than lambs in the Herb–Herb_EW_, Grass–Herb_D15EW_, Grass–Herb_EW_ and Grass–Grass_CW_ treatments that did not differ (*p* > 0.05).

In 2018, at L44 (date of early weaning), the live weights of lambs did not differ (*p* > 0.05) among the treatments (Table 3). At L61 and L87, Herb–Herb_CW_ and Grass–Herb_CW_ lambs were ~2 kg heavier (*p* < 0.05) than Herb–Herb_EW_, Grass–Herb_D15EW_ and Grass–Herb_EW_ lambs that did not differ (*p* > 0.05). Liveweight gain of Grass–Herb_CW_ lambs between L44 and L87 was greater (*p* < 0.05) than Herb–Herb_CW_ lambs, which in turn was greater (*p* < 0.05) than Herb–Herb_EW_, Grass–Herb_D15EW_ and Grass–Herb_EW_ lambs, which did not differ (*p* > 0.05).

### 3.3. Ewe Live Weight and Liveweight Gain

In 2017, at L51 (date of early weaning), the live weights of ewes did not differ (*p* > 0.05) between treatments (Table 4). At L65, the live weights of Herb–Herb_EW_, Herb–Herb_CW_, Grass–Herb_D15EW_, Grass–Herb_CW_ ewes did not differ (*p* > 0.05). Herb–Herb_EW_ ewes were 3 kg heavier (*p* < 0.05) than Grass–Herb_EW_ and Grass–Grass_CW_ ewes whose live weight did not differ (*p* > 0.05). At L99 (at conventional weaning), Herb–Herb_CW_, Herb–Herb_EW_, Grass–Herb_D15EW_, Grass–Herb_CW_ ewes did not differ (*p* > 0.05) in live weight but were 2 to 5 kg heavier (*p* < 0.05) than Grass–Herb_EW_ and Grass–Grass_CW_ ewes. Between L51 and L99, the liveweight gain of Herb–Herb_CW_, Herb–Herb_EW_, Grass–Herb_D15EW_, Grass–Herb_CW_ and Grass–Herb_EW_ ewes did not differ (*p* > 0.05) but were at least 30 g/day greater (*p* < 0.05) than Grass–Grass_CW_ ewes.

In 2018, at L44, L61 and L87, the live weights of ewes in all five treatments did not differ (*p* > 0.05) (Table 4). Similarly, the liveweight gain of ewes between L44 and L87 in each treatment did not differ (*p* > 0.05).

### 3.4. Ewe Body Condition Score

In 2017, at L51 (date of early weaning), the BCS of ewes did not differ (*p* > 0.05) between treatments (Table 5). At L65, the BCS of Herb–Herb_CW_, Herb–Herb_EW_, Grass–Herb_D15EW_ and Grass–Herb_EW_ ewes did not differ (*p* > 0.05). The BCS of Grass–Herb_CW_ and Grass–Grass_CW_ ewes did not differ (*p* > 0.05) but were lower (*p* < 0.05) than Herb–Herb_EW_, Grass–Herb_D15EW_ and Grass–Herb_EW_ ewes. At L99 (at conventional weaning), the BCS of Herb–Herb_CW_, Herb–Herb_EW_, Grass–Herb_D15EW_ and Grass–Herb_EW_ and Grass–Herb_CW_ were greater (*p* < 0.05) than Grass–Grass_CW_ ewes.

In 2018, at L44 (date of early weaning), the BCS of Herb–Herb_CW_, Herb–Herb_EW_ and Grass–Herb_D15EW_ ewes were greater (*p* < 0.05) than the Grass–Herb_CW_ and Grass–Herb_EW_ ewes that did not differ (*p* > 0.05). At L61, the BCS of Grass–Herb_CW_ ewes was lower (*p* < 0.05) than Herb–Herb_EW_ and Grass–Herb_D15EW_ ewes but was similar (*p* > 0.05) to Herb–Herb_CW_ and Grass–Herb_EW_ ewes. At L87 (at conventional weaning), the BCS of Herb–Herb_CW_, Herb–Herb_EW_, Grass–Herb_D15EW_ and Grass–Herb_EW_ ewes did not differ (*p* > 0.05). The BCS of Grass–Herb_CW_ ewes was lower (*p* < 0.05) than Grass–Herb_D15EW_ and Grass–Herb_EW_ ewes but was similar (*p* > 0.05) to that of Herb–Herb_CW_ and Herb–Herb_EW_ ewes.

### 3.5. Digestive Tract Components and Rumen Papillae Dimensions of Lambs in 2018

The weight of the empty digestive tract components of lambs between treatments at either L44 (at early weaning) or L87 (at conventional weaning) did not differ (*p* > 0.05; Table 6). At L44, papillae length and width of Grass–Herb_EW_ and Herb–Herb_EW_ lambs did not differ (*p* > 0.05; Table 6). At L87, Grass–Herb_EW_ and Herb–Herb_EW_ lambs had ~200 μm longer (*p* < 0.05) papillae than Grass–Herb_CW_ and Herb–Herb_CW_ lambs, however, the width of ruminal papillae did not differ (*p* > 0.05) between any of the treatments.

## 4. Discussion

The duration of the exposure of lambs to a herb–clover mix prior to early weaning, either from birth to early weaning, or from 15 days of age (L15) to early weaning, had no effect on lamb growth rates post-early weaning or their live weights at conventional weaning age, in either year of the current study. This suggests that this management option is not needed to improve the growth of lambs after early weaning onto herb–clover mix. This finding is supported by the limited differences in rumen measurements observed in 2018. In early life, lambs rely primarily on milk to fulfil their nutritional requirements and herbage consumption is low [33]. In the present study, lambs had ad-libitum access to their dams’ milk prior to early weaning, potentially limiting their need to consume herbage to fulfil their nutritional requirements thereby mitigating against any potential advantage from prolonged early exposure to the herb–clover mix. Combined, these results, and those from previous studies [9,10,11], indicate that a four-day adaptation period to the herb–clover mix, before early weaning, is all that is required to ensure adequate growth to a traditional weaning age.

Lambs weaned early onto a herb–clover mix grew at a similar rate, and were as heavy at conventional weaning, as unweaned lambs on a ryegrass–clover-based pasture in 2017. This was the only year that this later treatment was utilised. The growth rates achieved in the current study for early-weaned lambs on a herb–clover mix, were similar to those previously reported for commercially-reared-twin lambs on ryegrass white clover in New Zealand [34]. This suggests that lambs weaned at ~45 days of age and at a minimum live weight of 14 kg onto a herb–clover mix, have the potential to achieve liveweight gains similar to unweaned lambs reared on ryegrass–clover-based pasture. This adequate level of growth of lambs weaned early onto a herb–clover mix was likely driven by two potential factors. Firstly, increased herbage intake, due to herbs and clovers having lower fiber content and faster rumen flow through rates, than ryegrass [35]. Secondly, through greater preferential selection of different plant species [33] as lambs prefer herbs and clovers over ryegrass [36], or a combination of these variables. These results, in combination with previous studies [9,10,11] indicate early-weaning, onto a herb–clover mix is a management tool that can be used by farmers without risk of poor live weights at ~100 days of age.

In both years of the current study, lambs weaned early onto a herb–clover mix had lower liveweight gains to a conventional weaning age, than unweaned lambs on a herb–clover mix. This was likely due to unweaned lambs having greater total nutrient intake from both milk and herbage, compared to early-weaned lambs that had access only to herbage post-early weaning. Interestingly, the papillae length of early-weaned lambs was 200 μm greater than unweaned lambs on the herb–clover mix at conventional weaning age in 2018. The longer papillae of early-weaned lambs were likely driven by the need for greater dry matter intake to fulfil their nutritional requirements. Dry matter supplementation early in life is known to improve dry matter intake, rumination, the establishment of rumen microflora and alter feeding behaviors of neonatal ruminates [37]. Previously it has been shown that the development of the rumen, either reticulo-rumen capacity or the papillae length, of lambs did not differ at 90 days of age, even under conditions when differences in lamb growth and dry matter intake occurred [38]. That finding, combined with the present finding in regards to papillae length, may suggest that prior to day 90, any potential positive effects of rumen development on lamb growth may not occur and that studies should consider examining lambs for longer term impacts, post a traditional weaning age.

Lambs left unweaned on the herb–clover mix were 4 kg heavier at conventional weaning than those on ryegrass–clover-based pastures in 2017, supporting previous findings [9,10,11]. In both years of the study, their variations in herbage availability and quality were observed, although not at levels which would have restricted lamb intake [24,25]. Similarly, herbage quality and, in particular, ME and NDF were not inadequate to meet the requirements for lamb growth [39]. These parameters, therefore, do not explain why unweaned lamb growth was greater in the herb–clover mix treatment than unweaned lambs on ryegrass–clover-based pastures. As eluded to earlier, the likely explanation for the greater growth rate of unweaned lambs on herb–clover mix is the greater milk production of their dams, compared to those on ryegrass white clover [40], although the milk and herbage intake of lambs was not measured in this study. The results of this study, and previous studies [9,10,11] indicate grazing unweaned lambs on herb–clover mix is a management tool that farmers can utilize to increase lamb live weight at the conventional weaning age.

This study also allowed for the examination of the impact of moving ewes and lambs from a ryegrass pasture onto herb–clover mix at approximately 45 days of age without early weaning. The results in 2017 indicated that lambs were heavier and grew faster to conventional weaning than unweaned lambs that remained on ryegrass–clover-based pastures. Farmers, therefore, could consider moving ewes and lambs onto the herb–clover mix as an alternative to lambing ewes on a herb–clover mix. Further, unweaned lambs moved to a herb–clover mix at approximately 45 days of age grew at similar rates in 2017 and faster in 2018 than lambs reared with their dams on herb–clover mix from birth to conventional weaning. This suggests that farmers could use ryegrass–clover-based pasture for the lambing period and then move ewes and lambs to a herb–clover mix at 45 days of age to improve lamb weaning weights.

Ewes whose lambs were weaned early and offered ryegrass–clover-based pasture until conventional weaning had greater liveweight gains and BCS than unweaned ewes on ryegrass–clover-based pasture. This was likely due to the cessation of lactation, allowing ewes to partition nutrients to gaining live weight and was consistent with previous studies [10,11]. In both years, unweaned ewes on the herb–clover mix also had similar liveweight gains from early weaning to conventional weaning compared to ewes whose lambs were weaned early and offered ryegrass–clover-based pasture, matching previous study results [11]. Combined, these results suggest that ewes can either be weaned early onto ryegrass–clover-based pasture or left unweaned on herb–clover mix and achieve greater live weights than ewes on ryegrass–clover-based pasture with their lambs. Greater live weights at weaning can lead to greater live weights at mating which has been reported to have a positive impact on ewe reproductive performance [41]. Early weaning can be used as a technique to improve future ewe live weights and reproductive performance.

## 5. Conclusions

Early-weaned lambs on a herb–clover mix had similar liveweight gains to conventional weaning as lambs left unweaned on a ryegrass–clover-based pasture. Prolonged exposure of lambs to the herb–clover mix prior to early weaning, however, had no impact on their rumen development or subsequent growth, therefore, this management option cannot be used to improve performance post-early weaning. Ewes whose lambs were weaned early gained greater live weights and BCS compared to ewes which remained with their lambs on a ryegrass–clover-based pasture. Early weaning of lambs with a herb–clover mix is a management tool that farmers can use to achieve adequate growth in their lambs to a conventional weaning age whilst allowing their ewes to gain additional live weight. Greater lamb performance to conventional weaning, however, was achieved when lambs were left unweaned on the herb–clover mix.

## Figures and Tables

**Figure 1 animals-10-01354-f001:**
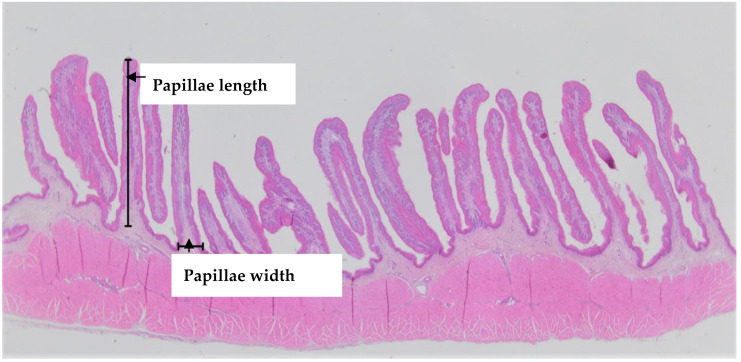
Example of histomorphometric measurements (papillae length and width) of the rumen of lambs measured using a computerised micrometre at 100× magnification.

**Figure 2 animals-10-01354-f002:**
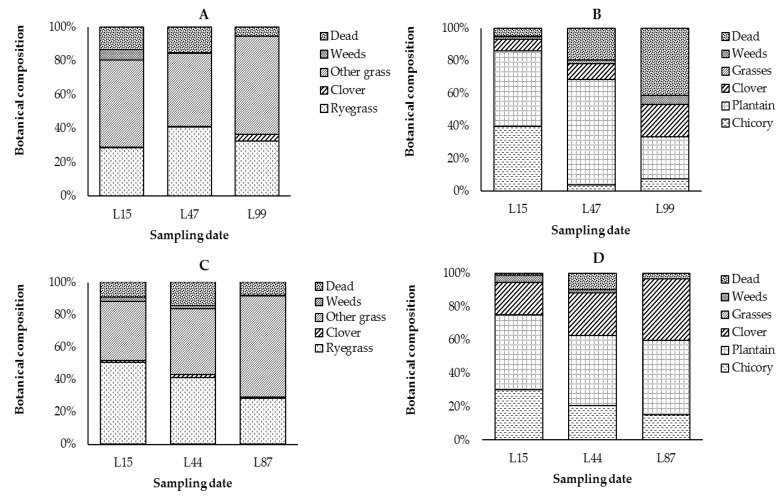
The botanical composition of the components of ryegrass–clover-based pasture (**A**), herb–clover mix (**B**) in 2017 and ryegrass–clover-based pasture (**C**) and herb–clover mix (**D**) in 2018 on 15, 51 and 99 days of age in 2017 and 2018 (L15, L44 and L87).

**Table 1 animals-10-01354-t001:** Summary of the experimental design with a description of treatments, year, number of lambs in each treatment, treatment herbage description and age of weaning.

Treatment	Year	Number of Lambs	Treatment Description	Lamb Age ^1^			
				0	15	47–51	87–99
Grass–Grass_CW_	2017	36	Lambs born and remained on ryegrass–clover-based pasture until conventional weaning	◆	◆	◆	◆,●
Grass–Herb_EW_	2017 2018	36 42	Lambs born on ryegrass–clover-based pasture and early weaned onto herb–clover mix	◆	◆	★,●	★
Grass–Herb_CW_	2017 2018	36 44	Lambs born on ryegrass–clover-based pasture, transferred with dam onto herb–clover mix at early weaning and remained until conventional weaning	◆	◆	★	★,●
Grass–Herb_D15EW_	2017 2018	30 42	Lambs born on ryegrass–clover-based pasture, transferred with dam onto herb–clover mix at 15 days of age and early weaned	◆	★	★,●	★
Herb–Herb_CW_	2017 2018	34 36	Lambs born and remained on herb–clover mix until conventional weaning	★	★	★	★,●
Herb–Herb_EW_	2017 2018	36 40	Lambs born on herb–clover mix and early weaned onto herb–clover mix	★	★	★,●	★

^1^ Management of the lambs during each time period. ◆ Lambs on ryegrass–clover-based pasture; ★ Lambs on herb–clover mix; ● Weaning age; Early weaned ewes were removed and grazed on ryegrass–clover-based pastures; early weaning occurred at L51 and L44 (average days of age) in 2017 and 2018, respectively; conventional weaning occurred at L99 and L87 days of age in 2017 and 2018, respectively. Grass–GrassCW, Grass–HerbEW and Grass–HerbCW lambs have been used in a previous study [11].

**Table 2 animals-10-01354-t002:** Herbage mass (HM) offered to both lambs and ewes, crude protein (CP), neutral detergent fibre (NDF) acid detergent fibre (ADF), dry matter digestibility (DMD) and metabolisable energy content (ME) of herbages collected L15, L51, L99 in 2017 and 2018 (L15, L44, L87) (least-squares mean ± SEM).

Year	Herbage Type	Study Day	HM (kg DM/ha)	CP (%)	NDF (%)	ADF (%)	DMD (%)	ME (MJ/Kg)
2017	Herb–clover mix	L15	2742 ^b,c^ ± 260	10.9 ^a^ ± 1.1	50.4 ^c,d^ ± 1.4	35.6 ^b^ ± 1.3	65.7 ^b^ ± 0.7	9.2 ^a,b^ ± 0.1
		L51	3221 ^c^ ± 260	17.3 ^b^ ± 1.1	46.7 ^c^ ± 1.4	26.6 ^a^ ± 1.3	65.8 ^b^ ± 0.7	9.5 ^b^ ± 0.1
		L99	3048 ^b,c^ ± 260	15.6 ^b^ ± 1.1	53.7 ^d^ ± 1.4	30.5 ^b^ ± 1.3	63.5 ^a^ ± 0.7	9.1 ^a^ ± 0.1
	Ryegrass–clover-based pasture	L15	2313 ^a,b^ ± 260	16.5 ^b^ ± 1.1	28.7 ^a^ ± 1.4	22.9 ^a^ ± 1.3	75.5 ^d^ ± 0.7	11.0 ^d^ ± 0.1
		L51	1899 ^a^ ± 260	12.0 ^a^ ± 1.1	32.7 ^b^ ± 1.4	26.2 ^a^ ± 1.3	72.0 ^c^ ± 0.7	10.6 ^c^ ± 0.1
		L99	2680 ^b^ ± 260	13.0 ^a,b^ ± 1.1	31.4 ^a,b^ ± 1.4	23.7 ^a^ ± 1.3	73.1 ^c^ ± 0.7	10.6 ^c^ ± 0.1
2018	Herb–clover mix	L15	4298 ^d^ ± 260	14.4 ^a^ ± 1.1	34.0 ^b^ ± 1.4	24.2 ^b^ ± 1.3	71.4 ^b,c^ ± 0.7	10.5 ^c,d^ ± 0.1
		L44	3481 ^c^ ± 260	17.2 ^b^ ± 1.1	27.6 ^a^ ± 1.4	19.5 ^a^ ± 1.3	74.1 ^d^ ± 0.7	11.0 ^e^ ± 0.1
		L87	3857 ^c^ ± 260	12.1 ^a^ ± 1.1	31.7 ^b^ ± 1.4	22.4 ^a,b^ ± 1.3	73.2 ^c,d^ ± 0.7	10.8 ^d,e^ ± 0.1
	Ryegrass–clover-based pasture	L15	1684 ^a^ ± 260	17.3 ^b^ ± 1.1	47.5 ^d^ ± 1.4	26.1 ^b^ ± 1.3	64.7 ^a^ ± 0.7	9.3 ^a^ ± 0.1
		L44	2057 ^a^ ± 260	12.9 ^a^ ± 1.1	48.0 ^d^ ± 1.4	24.6 ^b^ ± 1.3	65.9 ^a^ ± 0.7	9.8 ^b^ ± 0.1
		L87	2068 ^a^ ± 260	18.4 ^b^ ± 1.1	42.4 ^c^ ± 1.4	21.8 ^a,b^ ± 1.3	69.5 ^b^ ± 0.7	10.4 ^b,c^ ± 0.1

L, average days of age. ^a–e^ Means with different superscripts within columns are significantly different across years and treatments (*p* < 0.05).

**Table 3 animals-10-01354-t003:** Impact of weaning treatment; Herb–Herb_CW_, Herb–Herb_EW_, Grass–Herb_D15EW_, Grass–Herb_CW_, Grass–Herb_EW_, Grass–Grass_CW_ on live weight of lambs on L51, L65 and L99 in 2017 and L44, L61 and L87 in 2018; and liveweight gain (LWG) during L51–L99 in 2017 and during L44–L87 in 2018 (least-squares mean ± SEM).

Herbage Treatment	Lamb Live Weight (kg)						LWG (g/day)
	*n*		*n*		*n*		
2017							
	L51		L65		L99		L51–L99
Herb–Herb_CW_	42	19.3 ± 0.2 ^a^	41	23.5 ± 0.2 ^d^	42	34.7 ± 0.4 ^f^	317 ± 7 ^b^
Herb–Herb_EW_	44	19.4 ± 0.2 ^a^	43	21.9 ± 0.2 ^b^	43	31.1 ± 0.4 ^e^	244 ± 7 ^a^
Grass–Herb_D15EW_	38	19.1 ± 0.2 ^a^	38	21.7 ± 0.2 ^b^	38	30.1 ± 0.4 ^e^	231 ± 8 ^a^
Grass–Herb_CW_	42	19.3 ± 0.2 ^a^	41	23.1 ± 0.2 ^c,d^	42	34.6 ± 0.4 ^f^	318 ± 7 ^b^
Grass–Herb_EW_	46	19.0 ± 0.2 ^a^	45	21.6 ± 0.2 ^b^	46	30.6 ± 0.4 ^e^	239 ± 7 ^a^
Grass–Grass_CW_	34	19.2 ± 0.2 ^a^	33	22.7 ± 0.2 ^c^	34	30.3 ± 0.4 ^e^	231 ± 8 ^a^
2018							
	L44		L61		L87		L44–L87
Herb–Herb_CW_	36	19.0 ± 0.1 ^a^	36	25.0 ± 0.2 ^c^	35	31.7 ± 0.4 ^e^	289 ± 8 ^b^
Herb–Herb_EW_	36	18.9 ± 0.1 ^a^	34	23.0 ± 0.2 ^b^	33	28.9 ± 0.4 ^d^	232 ± 8 ^a^
Grass–Herb_D15EW_	42	18.9 ± 0.1 ^a^	41	22.7 ± 0.2 ^b^	41	28.7 ± 0.3 ^d^	231 ± 7 ^a^
Grass–Herb_CW_	34	18.8 ± 0.1 ^a^	34	24.8 ± 0.2 ^c^	34	32.1 ± 0.3 ^e^	321 ± 8 ^c^
Grass–Herb_EW_	34	18.8 ± 0.1 ^a^	34	22.6 ± 0.2 ^b^	34	28.7 ± 0.4 ^d^	237 ± 8 ^a^

L, average days of age; Herb–Herb_CW_, lambs born on herb–clover mix and conventional weaning (at ~99 days of age); Herb–Herb_EW_, lambs born on herb–clover mix and early weaned onto herb–clover mix at ~45 days of age; Grass–Herb_D15EW_, lambs born on ryegrass–clover-based pasture and transferred with dam onto herb–clover mix at 15 days of age and early weaned at ~45 days of age; Grass–Herb_CW_, lambs born on ryegrass–clover-based pasture and transferred with dam onto herb–clover mix at ~45 days of age and conventional weaning; Grass–Herb_EW_, lambs born on ryegrass–clover-based pasture and early weaned at ~45 days of age onto herb–clover mix; Grass–Grass_CW_, lambs born on ryegrass–clover-based pasture and conventional weaning. ^a–f^ Means with different superscripts are significantly different across years and treatments.

**Table 4 animals-10-01354-t004:** Impact of weaning treatment; Herb–Herb_CW_, Herb–Herb_EW_, Grass–Herb_D15EW_, Grass–Herb_CW_, Grass–Herb_EW_, Grass–Grass_CW_ on live weight of ewes on L51, L65 and L99 in 2017 and L44, L61 and L87 in 2018; and liveweight gain (LWG) during L51–L99 in 2017 and during L44–L87 in 2018 (least-squares mean ± SEM).

Weaning Treatment	Ewe Live Weight (kg)						LWG (g/day)
	*n*		*n*		*n*		
2017							
	L51		L65		L99		L51–L99
Herb–Herb_CW_	21	68.8 ± 0.9 ^a^	21	71.0 ± 0.9 ^a,b^	20	76.3 ± 1.0 ^b^	145 ± 18 ^b^
Herb–Herb_EW_	22	70.0 ± 0.8 ^a^	22	73.0 ± 0.8 ^b^	21	76.7 ± 0.9 ^b^	129 ± 17 ^b^
Grass–Herb_D15EW_	18	70.1 ± 0.9 ^a^	18	71.7 ± 1.0 ^a,b^	18	76.3 ± 1.0 ^b^	119 ± 19 ^b^
Grass–Herb_CW_	21	70.0 ± 0.9 ^a^	20	71.1 ± 0.9 ^a,b^	21	77.1 ± 0.9 ^b^	136 ± 17 ^b^
Grass–Herb_EW_	23	68.2 ± 0.9 ^a^	23	70.5 ± 0.9 ^a^	22	73.9 ± 0.9 ^a^	112 ± 17 ^b^
Grass–Grass_CW_	17	68.1 ± 0.9 ^a^	16	70.4 ± 1.0 ^a^	17	72.0 ± 1.0 ^a^	78 ± 19 ^a^
2018							
	L44		L61		L87		L44–L87
Herb–Herb_CW_	18	69.5 ± 1.0 ^a^	18	72.2 ± 1.0 ^a^	18	74.0 ± 1.0 ^a^	113 ± 19 ^a^
Herb–Herb_EW_	18	71.2 ± 0.9 ^a^	17	72.8 ± 1.0 ^a^	17	74.7 ± 1.0 ^a^	81 ± 19 ^a^
Grass–Herb_D15EW_	21	71.0 ± 0.9 ^a^	21	72.8 ± 0.9 ^a^	21	74.5 ± 0.9 ^a^	76 ± 18 ^a^
Grass–Herb_CW_	17	69.8 ± 0.8 ^a^	16	73.8 ± 0.8 ^a^	16	73.9 ± 0.9 ^a^	89 ± 17 ^a^
Grass–Herb_EW_	17	70.1 ± 0.9 ^a^	17	74.3 ± 0.9 ^a^	14	74.6 ± 1.0 ^a^	103 ± 19 ^a^

L, average days of age; Herb–Herb_CW_, lambs born on herb–clover mix and conventional weaning (at ~99 days of age); Herb–Herb_EW_, lambs born on herb–clover mix and early weaned onto herb–clover mix at ~45 days of age; Grass–Herb_D15EW_, lambs born on ryegrass–clover-based pasture and transferred with dam onto herb–clover mix at 15 days of age and early weaned at ~45 days of age; Grass–Herb_CW_, lambs born on ryegrass–clover-based pasture and transferred with dam onto herb–clover mix at ~45 days of age and conventional weaning; Grass–Herb_EW_, lambs born on ryegrass–clover-based pasture and early weaned at ~45 days of age onto herb–clover mix; Grass–Grass_CW_, lambs born on ryegrass–clover-based pasture and conventional weaning. ^a–b^ Means with different superscripts are significantly different across years and treatments.

**Table 5 animals-10-01354-t005:** Impact of weaning treatment; Herb–Herb_CW_, Herb–Herb_EW_, Grass–Herb_D15EW_, Grass–Herb_CW_, Grass–Herb_EW_, Grass–Grass_CW_ on the BCS of ewes at L51, L65 and L99 in 2017 and at L44, L61 and L87 in 2018 (results displayed as back transformed logit mean and 95% confidence interval).

Herbage Treatment	Ewe Body Condition Score					
	*n*		*n*		*n*	
2017						
	L51		L65		L99	
Herb–Herb_CW_	21	2.9 (2.5–2.9) ^a^	21	3.1 (3.1–3.6) ^a,b^	20	3.5 (3.1–3.4) ^b^
Herb–Herb_EW_	22	2.9 (2.8–3.1) ^a^	22	3.3 (3.1–3.6) ^b^	21	3.3 (3.1–3.6) ^b^
Grass–Herb_D15EW_	18	2.9 (2.7–3.1) ^a^	18	3.4 (3.2–3.7) ^b^	18	3.5 (3.3–3.7) ^b^
Grass–Herb_CW_	21	2.7 (2.5–2.9) ^a^	20	2.8 (2.6–3.1) ^a^	21	3.3 (3.1–3.4) ^b^
Grass–Herb_EW_	23	2.9 (2.8–3.1) ^a^	23	3.4 (3.1–3.6) ^b^	22	3.4 (3.2–3.5) ^b^
Grass–Grass_CW_	17	2.8 (2.6–3.0) ^a^	16	2.9 (2.2–3.0) ^a^	17	2.9 (2.7–3.1) ^a^
2018						
	L44		L61		L87	
Herb–Herb_CW_	18	2.5 (2.5–2.9) ^b^	18	2.8 (2.6–3.0) ^a,b^	18	3.1 (2.8–3.4) ^a,b^
Herb–Herb_EW_	18	2.9 (2.7–3.3) ^b^	17	3.0 (2.8–3.3) ^b^	17	3.1 (2.8–3.6) ^a,b^
Grass–Herb_D15EW_	21	2.9 (2.6–3.2) ^b^	21	3.0 (2.9–3.2) ^b^	21	3.4 (3.2–3.7) ^b^
Grass–Herb_CW_	18	2.4 (2.2–2.6) ^a^	16	2.6 (2.4–2.7) ^a^	16	2.8 (2.6–3.1) ^a^
Grass–Herb_EW_	17	2.3 (2.1–2.5) ^a^	17	2.9 (2.6–3.1) ^a,b^	14	3.3 (3.3–3.8) ^b^

L, average days of age; Herb–Herb_CW_, lambs born on herb–clover mix and conventional weaning (at ~99 days of age); Herb–Herb_EW_, lambs born on herb–clover mix and early weaned onto herb–clover mix at ~45 days of age; Grass–Herb_D15EW_, lambs born on ryegrass–clover-based pasture and transferred with dam onto herb–clover mix at 15 days of age and early weaned at ~45 days of age; Grass–Herb_CW_, lambs born on ryegrass–clover-based pasture and transferred with dam onto herb–clover mix at ~45 days of age and conventional weaning; Grass–Herb_EW_, lambs born on ryegrass–clover-based pasture and early weaned at ~45 days of age onto herb–clover mix; Grass–Grass_CW_, lambs born on ryegrass–clover-based pasture and conventional weaning. ^a-b^ Means with different superscripts are significantly different across years and treatments.

**Table 6 animals-10-01354-t006:** Mean weight of hot carcass, total digestive tract, reticulo-rumen, empty small intestine and empty large intestine and mean length and width of rumen papillae of Herb–Herb_EW_ and Grass–Herb_EW_ lambs at L44; and Grass–Herb_CW_, Herb–Herb_CW_, Herb–Herb_EW_ and Grass–Herb_EW_ lambs at L87 in 2018 (least-squares mean ± SEM).

Treatment		Hot Carcass Weight (kg)	Total Digestive Tract Weight (kg)	Reticulo-Rumen Weight (g)	Empty Small Intestine Weight (g)	Empty Large Intestine Weight (g)	Papillae Length (μm)	Papillae Width (μm)
Herb–Herb_EW_	L44	8.9 ± 0.3	3.5 ± 0.3	458 ± 40	458 ± 40	296 ± 51	809 ± 136	268 ± 16
Grass–Herb_EW_		9.6 ± 0.3	3.3 ± 0.3	410 ± 40	410 ± 40	234 ± 51	732 ± 148	260 ± 17
Grass–Herb_CW_	L87	14.7 ± 0.3	10.1 ± 0.3	951 ± 50	1353 ± 85	483 ± 80	892 ± 135 ^a^	221 ± 16
Herb–Herb_CW_		14.6 ± 0.3	10.3 ± 0.3	884 ± 53	1275 ± 90	425 ± 85	1090 ± 152 ^a^	211 ± 18
Herb–Herb_EW_		14.3 ± 0.3	10.8 ± 0.3	1013 ± 51	1119 ± 86	611 ± 80	1225 ± 137 ^b^	242 ± 16
Grass–Herb_EW_		14.0 ± 0.3	10.9 ± 0.3	970 ± 55	1280 ± 93	463 ± 87	1333 ± 122 ^b^	245 ± 14

L, average days of age; Herb–Herb_EW_, lambs born on herb–clover mix and early weaned onto herb–clover mix at ~45 days of age; Grass–Herb_EW_, lambs born on ryegrass–clover-based pasture and early weaned at ~45 days of age onto herb–clover mix; Grass–Herb_CW_, lambs born on ryegrass–clover-based pasture and transferred with dam onto herb–clover mix at ~45 days of age and conventional weaning; Herb–Herb_CW_, lambs born on herb–clover mix and conventional weaning (at ~99 days of age). ^a,b^ Means with different superscripts are significantly different across treatments.

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
