# Peer review of "Pre-Exposure of Early-Weaned Lambs to a Herb-Clover Mix Does Not Improve Their Subsequent Growth"

_animals, 2020, doi:10.3390/ani10081354_

Round 1

Reviewer 1 Report

The authors attempt to identify the effect of feeding herb-clover mix on the subsequent growth rate of kids. The study is robust in its design. However, the parameters measured are not sufficient to test the hypothesis mentioned in the study. The earlier study in previous years by the authors has already explained that body weight do not vary in kids exposed to herbs (Ekanayake, L.J.; Corner-Thomas, R.A.; Cranston, L.M.; Kenyon, P.R.; Morris, S.T. Lambs Weaned Early onto a Herb-Clover Mix Have the Potential to Grow at a Similar Rate to Unweaned Lambs on a Grass-472 Predominant Pasture. Animals. 2020, 10, 613). Additional parameters studied-body condition score and digestive tract parameters- provide no additional information on herb supplementation. This was possibly due to the fact that the experimental design was not sufficient to test the hypothesis. The previously used and the additional parameters measured had poor relationship with the response variables. For example, the authors explain that there is poor relationship between rumen development and growth of lambs in early life. However, in the manuscript, they test the relationship with the same parameters without any further modifications in the experimental design.

Abstract

Include significant results in the abstract. Specifically the changes in body condition score and observations on digestive tract changes need to be incorporated.

Materials and methods

Format table 1, 1st row

Table 1, give separate markings for ryegrass-clover based pasture and herb-clover mix, so that it is easy to identify the treatment groups at different pastures.

The experimental design was not sufficient to test the hypothesis. The parameters measured had poor relationship with the response variables. For example, the authors opine that there is poor relationship between rumen development and growth of lambs in early life. However, in the manuscript, they test the relationship with same parameters.

134-136- It is hypothesised that when kids are early weaned and kept with other dams, they wont suckle milk from other dams. What precautions were taken to ensure that those kids did not suckle other dams? If the kids suckled milk from other dams, it may have affected the whole results.

Results

Highlight the specific results and its significance with the other parameters studied in the experiment.

229-260- Botanical composition, herbage mass and nutritional quality of herbage varied significantly in all treatments between different days. The inconsistent results may be due to these variations in nutritional quality of herbage.

Discussion

Discuss the significant results of the study.

Author Response

Open Review

Authors appreciate the comments from the reviewers and the editor. Relevant changes have been made in the revised manuscript and author’s comments have been made for each point raised as follows.

English language and style

( ) Extensive editing of English language and style required 
( ) Moderate English changes required 
(x) English language and style are fine/minor spell check required 
( ) I don't feel qualified to judge about the English language and style 

Yes

Can be improved

Must be improved

Not applicable

Does the introduction provide sufficient background and include all relevant references?

( )

( )

(x)

( )

Is the research design appropriate?

( )

( )

(x)

( )

Are the methods adequately described?

( )

(x)

( )

( )

Are the results clearly presented?

( )

( )

(x)

( )

Are the conclusions supported by the results?

( )

( )

(x)

( )

Comments and Suggestions for Authors

The authors attempt to identify the effect of feeding herb-clover mix on the subsequent growth rate of kids. The study is robust in its design. However, the parameters measured are not sufficient to test the hypothesis mentioned in the study. The earlier study in previous years by the authors has already explained that body weight do not vary in kids exposed to herbs (Ekanayake, L.J.; Corner-Thomas, R.A.; Cranston, L.M.; Kenyon, P.R.; Morris, S.T. Lambs Weaned Early onto a Herb-Clover Mix Have the Potential to Grow at a Similar Rate to Unweaned Lambs on a Grass-472 Predominant Pasture. Animals2020, 10, 613). Additional parameters studied-body condition score and digestive tract parameters- provide no additional information on herb supplementation. This was possibly due to the fact that the experimental design was not sufficient to test the hypothesis. The previously used and the additional parameters measured had poor relationship with the response variables. For example, the authors explain that there is poor relationship between rumen development and growth of lambs in early life. However, in the manuscript, they test the relationship with the same parameters without any further modifications in the experimental design.

Authors comment: The literature indicates the reason for the variation in lamb growth (note it is lambs not kids as reviewer suggested) post-early weaning on a herb-clover mix, compared to unweaned lambs on a ryegrass-clover based pasture, is as yet unknown (see lines 45 to 48 in introduction) and not due to variation in herbage metabolisable energy or crude protein content. Therefore, it is unknown. We set out to determine if prior exposure to the herb mix would result in a more consistent lamb growth response post early weaning. Previous studies have suggested that early exposure to herbage-based diets can help develop a strong host microbiota in the rumen and experience with a herbage type can influence a lamb’s preference for that herbage later in life (see lines 73 to 74 in introduction). Further, the rumen development of early-weaned lambs offered herb-clover mix has not been studied to date, therefore, this relationship was tested in this study. More information has been added to the introduction to clarify this (Lines 78-79).

The design of this study is different from any of the previous early weaning studies reported in the literature with the herb-clover mix and is not the same as suggested by this reviewer. The other two reviewers did not comment on the hypothesis. We believe therefore the design does allow for testing of the hypothesis as we had early weaned lambs that did or did not have prior exposure to the herb- clover mix before early weaning. Therefore no change to the hypothesis has been made but additional information has been added to introduction to clarify.

Abstract

Include significant results in the abstract. Specifically the changes in body condition score and observations on digestive tract changes need to be incorporated.

Authors comment: More results have been added to the abstract in the revised manuscript. We have limited these to lamb growth and digestive tract weight due to the focus of the study (and the hypothesis) being the lambs not their dams. Adding dam information is not novel and may confuse the reader (see lines 32-39).

Materials and methods

Format table 1, 1st row

Authors comment: This has been corrected in the revised manuscript (Lines 121-123)

Table 1, give separate markings for ryegrass-clover based pasture and herb-clover mix, so that it is easy to identify the treatment groups at different pastures.

Authors comment: Separate symbols have been added to the table in the revised manuscript (Lines 121-123)

The experimental design was not sufficient to test the hypothesis. The parameters measured had poor relationship with the response variables. For example, the authors opine that there is poor relationship between rumen development and growth of lambs in early life. However, in the manuscript, they test the relationship with same parameters.

Authors comment: We disagree with this comment and neither of the other two reviewers had any issue with the hypothesis. The reviewer also seems to have missed that the hypothesis (see lines 81 to 84 in introduction) was focused firstly on lamb growth then secondly rumen development (not the other way around). The design had lambs which did or did not have prior exposure to the herb-clover mix prior to early weaning and growth and rumen traits were measured.

As indicated above, to date the literature has failed to explain the variation in lamb growth post early weaning. The literature had not examined the impact of early exposure to the herb clover mix on either lamb growth or rumen development.   We have altered aspects of the introduction to make these points clearer (see changes in the introduction) and have made it clearer in abstract and summary the lamb growth was the main aspect of the study. Further we have made it clearer in the discussion (see lines 412 to 418) that any difference in rumen development before a traditional weaning age, while present may not in fact alter lamb growth, and that longer term studies (post a traditional weaning age) are likely needed to see the potential advantage express itself.

134-136- It is hypothesised that when kids are early weaned and kept with other dams, they wont suckle milk from other dams. What precautions were taken to ensure that those kids did not suckle other dams? If the kids suckled milk from other dams, it may have affected the whole results.

Authors comment: We did not hypothesis they will not suckle that is not the hypothesis being tested. It is known they will not suckle. Within 1 to 2 h of birth, ewes develop an exclusive bond with their lambs (note again it is not kids as reviewer keeps suggesting) and reject any alien lamb that attempts to suck for the remainder of the lactation. It is well established in sheep that post this short few hours (let alone many weeks as in this study) a ewe will not allow alien lambs to suckle. Therefore, there was no concern regarding the early-weaned lambs suckling from unweaned ewes. Further, this method has been used in previously published articles in the journal of Animals (Ex: Ekanayake, L.J.; Corner-Thomas, R.A.; Cranston, L.M.; Kenyon, P.R.; Morris, S.T. Lambs Weaned Early onto a Herb-Clover Mix Have the Potential to Grow at a Similar Rate to Unweaned Lambs on a Grass-472 Predominant Pasture. Animals. 2020, 10, 613) and other publications by our group (Ekanayake, W.E.M.L.J.; Corner-Thomas, R.A.; Cranston, L.M.; Kenyon, P.R.; Morris, S.T. The effect of live weight at weaning on liveweight gain of early weaned lambs onto a herb-clover mixed sward. Proc. N. Z. Soc. Anim. Prod. 2017, 77, 37–42., Ekanayake, W.E.M.L.J; Corner-Thomas, R.A.; Cranston, L.M.; Kenyon, P.R.; Morris, S.T. A comparison of liveweight gain of lambs weaned early onto a herb-clover mixed sward and weaned conventionally onto a ryegrass-clover pasture and herb-clover mixed sward. Asian-Australas J Anim Sci. 2019, 32, 201–208., Ekanayake, L.J.; Corner-Thomas, R.A.; Cranston, L.M.; Kenyon, P.R.; Morris, S.T. Lambs Weaned Early onto a Herb-Clover Mix Have the Potential to Grow at a Similar Rate to Unweaned Lambs on a Grass-Predominant Pasture. Animals. 2020, 10, 613.) and in all of those studies we have also not observed any ‘stealing’ of milk post early weaning.

Results

Highlight the specific results and its significance with the other parameters studied in the experiment.

229-260- Botanical composition, herbage mass and nutritional quality of herbage varied significantly in all treatments between different days. The inconsistent results may be due to these variations in nutritional quality of herbage.

Authors comment: In both years there were observed variations in herbage availability and quality although not at levels which would have restricted lamb intake nor was the herbage quality, particularly ME and NDF, inadequate to meet the requirements for lamb growth. These parameters, therefore, do not explain why unweaned lamb growth was greater on the herb-clover mix treatment than unweaned lambs on ryegrass-clover based pastures (as has also been shown not to be the case in the other studies by our group – Ekanayake, W.E.M.L.J.; Corner-Thomas, R.A.; Cranston, L.M.; Kenyon, P.R.; Morris, S.T. The effect of live weight at weaning on liveweight gain of early weaned lambs onto a herb-clover mixed sward. Proc. N. Z. Soc. Anim. Prod. 2017, 77, 37–42., Ekanayake, W.E.M.L.J; Corner-Thomas, R.A.; Cranston, L.M.; Kenyon, P.R.; Morris, S.T. A comparison of liveweight gain of lambs weaned early onto a herb-clover mixed sward and weaned conventionally onto a ryegrass-clover pasture and herb-clover mixed sward. Asian-Australas J Anim Sci. 2019, 32, 201–208., Ekanayake, L.J.; Corner-Thomas, R.A.; Cranston, L.M.; Kenyon, P.R.; Morris, S.T. Lambs Weaned Early onto a Herb-Clover Mix Have the Potential to Grow at a Similar Rate to Unweaned Lambs on a Grass-Predominant Pasture. Animals. 2020, 10, 613.) and in all of those studies we have also not observed any ‘stealing’ of milk post early weaning). Remember also the design of these studies places both weaned and unweaned lambs together on the same herbage paddock, removing a potential paddock affect variable.

Discussion

Discuss the significant results of the study.

Authors comment: We are unsure what is meant by this comment as the paper does solely focus on the main points in the discussion. The other two reviewers did not comment on the discussion and we have made changes in line with their suggestions.

Note author S.T. Morris, a native English speaker, has thoroughly checked the manuscript for English.

Submission Date

10 June 2020

Date of this review

14 Jun 2020 10:25:58

Reviewer 2 Report

This study looking at the effect of exposing lambs to herb-clover prior to weaning is quiet a difficult study to discribe but the authors have done well in outlining the methods. It is resonable well written but I have several comments and concerns.

Title: The title needs to be shorter and to the point I suggest something like "Pre-exposure to a herb-clover mix does not improve preformance in early weaned lambs" 

L31-33: remove "it is, therefore" also suggest restructuring the sentence.

Introduction:

I think the introduction needs more justification for the reason of this study. This study appears to have very similar methods as those previsously written by the authors and I am not convinced as to what new information this study brings compared to previous work. 

L38: "allowing" also suggest spliting sentence in half.

L58 & L60: remove "therefore" both are unnecessary

Methods and Materials:

L87: Why were twins only used in these studies and previous studies? Would being a twin not affect the growth rates of these lambs? Are there any other effects that may have  occured due to not sellecting singlton. Also what was the % of twin bearing ewes in these flocks compared to single bearing?

Table 1: This table is quite cluttered and difficult to interpret, treatment description in the table should be sufficient especially as you repeat in text at what ages, things occured as well as repeating the numbers of animals again in text. 

L108: Were the 2017 lambs the same lambs used in your previous study "Lambs Weaned Early onto a Herb-Clover Mix Have the Potential to Grow at a Similar Rate to Unweaned Lambs on a Grass-Predominant Pasture"

L117: Why were lambs moved from grass to herb at day 15? Also would it have been beneficial to also have done this with a conventially weaned Grass-Herb group?

L143: Why a disparity in numbers when there are even groups of CW and EW?

L144: "Early weaning" repeated

L153: skins "were" removed, delete repeating comma

L164: Alcohol "70, 95%" which is it

L181-2: Unclear as to what "dried to a constant weight" is

Results:

Table 2 is hard to read, the results are formatted to closely. Which are the most important measures to present? Maybe remove a few if possible. 

3.2. Is it possible to cut down on word count as results are presented in the table

L264: remove "in"

L266 and 272: The presentation of L99 (conventional weaning) L44 (date of early weaning) is used through out the text and makes it quite hard to follow as well as the use of these acronyms of L44, L61 etc. The authors flip between the description of L days after midpoint of lambing and just days of age. I suggest just using days of age and refering to it in text as "conventional weaning age" and "early weaning age". This will just need to be described in the methods once and will assist in cutting down on word count and improving the flow of text. 

3.3. Ewe live weight and gain: There is a large presentation of results on ewe live weight  and condition even though this was not the aim of this study and has previously been shown in your work. Although it may be important to report the ewes weight to ensure that they did not influence lamb growth, I do not find this section necessary to the study in question, either remove this section or cut down on the results as it is not the focus of the study.

Discussion:

L327: Remove "therefore"

L371-374: Reword

L382: Remove "therefore"

L392 - 394: Was this recorde in this study?

Page 12: excessive use of "therefore" please edit text to remove, unneeded word

Author Response

Open Review

Authors appreciate the comments from the reviewers and the editor. Relevant changes have been made in the revised manuscript and author’s comments have been made for each point raised as follows.

English language and style

( ) Extensive editing of English language and style required 
( ) Moderate English changes required 
(x) English language and style are fine/minor spell check required 
( ) I don't feel qualified to judge about the English language and style 

Yes

Can be improved

Must be improved

Not applicable

Does the introduction provide sufficient background and include all relevant references?

( )

(x)

( )

( )

Is the research design appropriate?

( )

(x)

( )

( )

Are the methods adequately described?

( )

(x)

( )

( )

Are the results clearly presented?

( )

(x)

( )

( )

Are the conclusions supported by the results?

(x)

( )

( )

( )

Comments and Suggestions for Authors

This study looking at the effect of exposing lambs to herb-clover prior to weaning is quiet a difficult study to discribe but the authors have done well in outlining the methods. It is resonable well written but I have several comments and concerns.

Title: The title needs to be shorter and to the point I suggest something like "Pre-exposure to a herb-clover mix does not improve preformance in early weaned lambs" 

Authors comment: title has been changed in the revised manuscript (Lines 2-5)

L31-33: remove "it is, therefore" also suggest restructuring the sentence.

Authors comment: The sentence has been restructured in the revised manuscript (Lines 43-45)

Introduction:

I think the introduction needs more justification for the reason of this study. This study appears to have very similar methods as those previsously written by the authors and I am not convinced as to what new information this study brings compared to previous work. 

Authors comment: As indicated to response to reviewer one this is the first study examining early weaning on the herb-clover mix where lambs have either had prior exposure or no prior exposure to the herb-clover mix and it is the first study to examine impacts on the lambs rumen at a young age. So the study is very different from previous work in this area. However, we accept that we were not as clear with regards to this in the introduction as we could. Hence changes have been made to the introduction (see lines 45 to 47, 64 to 67, 78 to 79 etc.)

L38: "allowing" also suggest spliting sentence in half.

Authors comment: Sentence has been restructured in the revised manuscript (Lines 43-45)

L58 & L60: remove "therefore" both are unnecessary

Authors comment: This has been removed in the revised manuscript (Lines 79-81)

Methods and Materials:

L87: Why were twins only used in these studies and previous studies?

Authors comment: Only twin lambs have been used in this study for following reasons

  1. Lambs born in New Zealand to mature ewes are predominantly twins, therefore, the focus of this study was the drivers of the growth of twin lambs, rather than singletons or triplets.
  2. The growth of singletons, twins and triplets are different due to several reasons including initial body weight and the amount of milk received from dams. If singletons and triplets were to be included in the study, it would have been difficult to make inferences due to these variations.
  3. In mid to late lactation the breeds of sheep utilised in NZ can struggle to produce enough milk for their lambs to support high growth rates, especially when herbage quality and quantity are limiting.
  4. The majority of the previous studies (Ekanayake, W.E.M.L.J.; Corner-Thomas, R.A.; Cranston, L.M.; Kenyon, P.R.; Morris, S.T. The effect of live weight at weaning on liveweight gain of early weaned lambs onto a herb-clover mixed sward. Proc. N. Z. Soc. Anim. Prod. 2017, 77, 37–42., Ekanayake, W.E.M.L.J; Corner-Thomas, R.A.; Cranston, L.M.; Kenyon, P.R.; Morris, S.T. A comparison of liveweight gain of lambs weaned early onto a herb-clover mixed sward and weaned conventionally onto a ryegrass-clover pasture and herb-clover mixed sward. Asian-Australas J Anim Sci. 2019, 32, 201–208., Ekanayake, L.J.; Corner-Thomas, R.A.; Cranston, L.M.; Kenyon, P.R.; Morris, S.T. Lambs Weaned Early onto a Herb-Clover Mix Have the Potential to Grow at a Similar Rate to Unweaned Lambs on a Grass-Predominant Pasture. Animals. 2020, 10, 613.) Examining the impacts of early weaning on the herb-clover mix with mature ewes have utilised twin rearing mature ewes.

Would being a twin not affect the growth rates of these lambs?

Authors comment: In this study only twins were utilised so the treatments are consistent in this regard. As mentioned above, on average twins grow slower than singletons. However, that is not being tested what is being tested is that if twins were early weaned onto the herb-clover herbage what is the impact on their growth and how is this impacted by prior early exposure to the herb-clover mix. Hence we believe no change is required to the manuscript.

Are there any other effects that may have occurred due to not selecting singleton?

  1. Authors comment: This study focused on twins only. But in a previous study with singleton lambs (Corner-Thomas, R.A., Cranston, L.M., Kemp, P.D., Morris, S.T., Kenyon, P.R. 2018. The performance of single-rearing ewes and their lambs offered ryegrass pasture or herb–clover mix during lactation. New Zealand Journal of Agricultural Research, 61, 67-80) it was found that early weaning on the herb-clove mix can improve lamb growth post early weaning. Hence we have not made any change to the manuscript.

Also what was the % of twin bearing ewes in these flocks compared to single bearing?

Authors comment: This question has no relevance to the design of the study which focused on twins only. However, the flock has a lambing percentage (ie number of foetuses per ewe at pregnancy diagnosis) of 1.8 to 1.9 clearly indicating twins are the majority. No change was thus made to the manuscript.

Table 1: This table is quite cluttered and difficult to interpret, treatment description in the table should be sufficient especially as you repeat in text at what ages, things occurred as well as repeating the numbers of animals again in text. 

Authors comment: For some unknown reason there were some symbols missing from the table, perhaps it happened during the conversion of word file to a pdf file, new symbols have been added to make it clear (Lines 121-123)

L108: Were the 2017 lambs the same lambs used in your previous study "Lambs Weaned Early onto a Herb-Clover Mix Have the Potential to Grow at a Similar Rate to Unweaned Lambs on a Grass-Predominant Pasture"

Authors comment: Lambs in three of the treatment groups (Grass-GrassCW, Grass-HerbEW, Grass-HerbCW) were used in the previous study ("Lambs Weaned Early onto a Herb-Clover Mix Have the Potential to Grow at a Similar Rate to Unweaned Lambs on a Grass-Predominant Pasture") as control groups. The other three 2017 treatments were not. The lambs in the 2018 groups (regardless of treatment) were not used.

L117: Why were lambs moved from grass to herb at day 15? Also would it have been beneficial to also have done this with a conventially weaned Grass-Herb group?

Authors comment: Lambs in the Grass-HerbD15EW were only transferred to herb-clover mix at 15 days of age to give them a pre-exposure prior to early weaning. Due to needing time to bond and the ability to move young lambs form paddocks it was not practical to do this earlier. One of the hypotheses being tested was whether early exposure improved the response in terms of lamb growth from early weaning. Therefore, transferring lambs in Grass-HerbCW (which were not early weaned) at day 15 was not necessary.

L143: Why a disparity in numbers when there are even groups of CW and EW?

Authors comment:

Firstly, histological examination of the digestive system at L44 was done to estimate the effect of feed type prior to early weaning on their rumen development. At this stage, lambs were either on grass-clover pasture or herb-clover mix, therefore, only 12 lambs were sacrificed (6 from grass-clover based pasture and 6 from herb-clover mix).

At L87, after early weaning treatments had occurred, the effect of weaning treatment and herbage type on the rumen development was estimated which needed to sacrifice 6 from each treatment (6 by 4=24), which resulted in this disparity.

The second comment is in regards to lambs recorded to weaning and differences in treatment numbers (see lines 129 to 152). This difference comes about because we only utilised (different post early weaning dates in 2017 and 2018 respectively) full alive sets of twins and those that were of the minimum live weight (which was an animal ethics and welfare requirement) when lambs were early weaned. While treatment allocation occurred just prior to lambing (see lines 107 to 108 in methods).

L144: "Early weaning" repeated

Authors comment: This has been corrected in the revised manuscript (Line 168)

L153: skins "were" removed, delete repeating comma

Authors comment: This has been corrected in the revised manuscript (Line 177)

L164: Alcohol "70, 95%" which is it

Authors comment: This has been corrected in the revised manuscript (Line 188)

L181-2: Unclear as to what "dried to a constant weight" is

Authors comment: Samples were dried until a constant weight was achieved

Results:

Table 2 is hard to read, the results are formatted to closely. Which are the most important measures to present? Maybe remove a few if possible. 

Authors comment: These parameters provide a complete description of the nutritional quality of both herbages. The other reviewers did not comment on this table and its clarity. Therefore, the authors wish for it to remain in the revised manuscript. However, if the editor requires, we could remove a couple of the variables outlined but then the data on the herbage will be less complete.

Table 3.2. Is it possible to cut down on word count as results are presented in the table

We have not reduced the results description of this table as we believe it needs to be accurate and the other reviewer did not require this. We could however if the editor wishes.

L264: remove "in"

Authors comment: This has been corrected in the revised manuscript (Line 290)

L266 and 272: The presentation of L99 (conventional weaning) L44 (date of early weaning) is used through out the text and makes it quite hard to follow as well as the use of these acronyms of L44, L61 etc. The authors flip between the description of L days after midpoint of lambing and just days of age. I suggest just using days of age and refering to it in text as "conventional weaning age" and "early weaning age". This will just need to be described in the methods once and will assist in cutting down on word count and improving the flow of text. 

Authors comment: The description of L, days after midpoint of lambing, and just days of age has been corrected to days of age in the revised manuscript.

The use of L, however, helps reduce the number of words immensely, especially when a list of dates are being stated i.e. L15, L61 etc. Therefore, authors agree to keep it in the manuscript (Line 129-131)

3.3. Ewe live weight and gain: There is a large presentation of results on ewe live weight and condition even though this was not the aim of this study and has previously been shown in your work. Although it may be important to report the ewes weight to ensure that they did not influence lamb growth, I do not find this section necessary to the study in question, either remove this section or cut down on the results as it is not the focus of the study.

Authors comment: It is important for completeness to examine the impacts of the treatments not only on the lambs but their dams. Therefore while we agree the hypothesis was focused on the lambs we believe the dam data should be shown and we wish for it to remain (note the other two authors did not suggest it should be removed). This has been the approach in all of our previous studies in this series.(Ekanayake et al 2017, 2019, 2020)

Discussion:

L327: Remove "therefore"

Authors comment: Authors finds no ‘therefore’ to remove in L327. However, we have removed non-needed ones (see lines 398-400).

L371-374: Reword

Authors comment: This has been corrected in the revised manuscript (Line new lines 398 - 403)

L382: Remove "therefore"

Authors comment: This has been corrected in the revised manuscript (see Line 417).

L392 - 394: Was this recorded in this study?

Authors comment: This has been corrected in the revised manuscript (see Line 427-429).

Page 12: excessive use of "therefore" please edit text to remove, unneeded word

Authors comment: This has been corrected with the removal of these through-out the revised manuscript

Submission Date

10 June 2020

Date of this review

26 Jun 2020 08:20:10

Reviewer 3 Report

Review for the manuscript entitled “Impact of early exposure of lambs to a herb-clover mix prior to early weaning on their subsequent growth”

The manuscript investigated the effect of early weaning of lambs along with prolonged early exposure to a herb-mix herbage on their liveweight gain and rumen development until the age of conventional weaning    

Overall, the manuscript is well written and presented, but few changes could make this manuscript better. In the view of this reviewer, the use of only herb-mix after early weaning (i.e., lack of Grass-Grass EW) is not convincing and has not been justified in the introduction. The authors mentioned studies in which lambs weaned early onto herb-mix and were compared to unweaned lambs on RG-clover based herbage, but why those lambs were not compared to early weaning onto RG based herbage?. I see two factors here: weaning time (early vs. Conventional) and herbage type (herbs vs. RG-clover), but the lack of “Grass-grass early weaning” treatment made it hard to isolate the effect of each factor. The question “why lambs were early weaned onto herb-mix and not RG-clover?” should be answered in the introduction.

The Herbage chemical and botanical composition data were not used in the discussion; although, could have been used to explain differences shown in the performance of lambs and ewes among treatments. I noticed in table 2 that herbage quality (i.e., the high ME and low NDF content) was greater in the RG-clover than herb-mix in 2017, but the opposite was true in 2018. This, and potential differences in DM intake, due to differences in HM, could have been discussed.

Specific comments

Line 22 explain conventional weaning

Line 30 what about the rest of the treatments? more results should be included in the abstract

Line 37-40 very long sentence consider a re-write

Line 42 delete “however”

Line 43-48 why was not compared to “grass-grass early weaning”?

Line 48 what is the primary driver then?

Table 1 I found this table confusing! Perhaps the symbol for “Lambs on herb-clover mix” is missing in the pdf review version? Check this please

Line 119 replace “onset of the main experiment” with “early weaning” or similar description

Line 153 delete repeated commas

Line 182 were these samples cut above ground level or expected post-grazing mass?

Line 242 oh! This is not true! Table 2 shows the opposite

Table 2 I understand that some of the weaned ewes joined the commercial flock. So, are these data in table 2 showing what the lambs grazed? What ewes grazed? Or an average of what offered for both

Line 264 delete “in”

Line 296 delete “which”

Line 379-382 any explanation for this? Intake is higher? ME content was even higher for grass than the herb in 2017!

Author Response

Open Review

Authors appreciate the comments from the reviewers and the editor. Relevant changes have been made in the revised manuscript and author’s comments have been made for each point raised as follows.

English language and style

( ) Extensive editing of English language and style required 
( ) Moderate English changes required 
(x) English language and style are fine/minor spell check required 
( ) I don't feel qualified to judge about the English language and style 

Yes

Can be improved

Must be improved

Not applicable

Does the introduction provide sufficient background and include all relevant references?

( )

(x)

( )

( )

Is the research design appropriate?

(x)

( )

( )

( )

Are the methods adequately described?

(x)

( )

( )

( )

Are the results clearly presented?

( )

(x)

( )

( )

Are the conclusions supported by the results?

(x)

( )

( )

( )

Comments and Suggestions for Authors

Review for the manuscript entitled “Impact of early exposure of lambs to a herb-clover mix prior to early weaning on their subsequent growth”

Reviewer 3.

Overall, the manuscript is well written and presented, but few changes could make this manuscript better. In the view of this reviewer, the use of only herb-mix after early weaning (i.e., lack of Grass-Grass EW) is not convincing and has not been justified in the introduction. The authors mentioned studies in which lambs weaned early onto herb-mix and were compared to unweaned lambs on RG-clover based herbage, but why those lambs were not compared to early weaning onto RG based herbage?. I see two factors here: weaning time (early vs. Conventional) and herbage type (herbs vs. RG-clover), but the lack of “Grass-grass early weaning” treatment made it hard to isolate the effect of each factor. The question “why lambs were early weaned onto herb-mix and not RG-clover?” should be answered in the introduction.

Authors comment: Lambs weaned early at 4-8 weeks of age onto a ryegrass-clover based pasture have consistently displayed lower live weight gain and poorer survival compared to unweaned lambs on ryegrass-clover based pastures to approximately 14 weeks of age (Mulvaney et al. 2009; Mulvaney et al. 2011). These suggest that early weaning lambs onto a grass-clover based pasture is not a suitable option for farmers and in fact are not utilised by them. Therefore, this treatment was not included in the study and could have had negative welfare for the lambs.

Further there is substantial literature now showing herb-clover mixes improve lambs growth post a traditional weaning age (see lines 48 to 53 in Introduction). This has resulted in increased interest to use these mixes for early weaning. That is why these series of studies have been undertaken.

The Herbage chemical and botanical composition data were not used in the discussion; although, could have been used to explain differences shown in the performance of lambs and ewes among treatments. I noticed in table 2 that herbage quality (i.e., the high ME and low NDF content) was greater in the RG-clover than herb-mix in 2017, but the opposite was true in 2018. This, and potential differences in DM intake, due to differences in HM, could have been discussed.

Authors comment: Previous studies (Ekanayake et al. 2017, 2019, 2020) have concluded that the variation of nutritional quality of herbages had no effect on lambs performance in previous studies in this series. In both years there were observed variations in herbage availability and quality although not at levels which would have restricted lamb intake nor was the herbage quality, particularly ME and NDF, inadequate (or excessive) to meet or restrict the nutritional requirements for lamb growth. Note we do discuss the herbage quality see in the discussion see Lines 447-455 and we do not believe further discussion is needed.

Specific comments

Line 22 explain conventional weaning

Authors comment: These changes have been made in the revised manuscript (Line 25)

Line 30 what about the rest of the treatments? more results should be included in the abstract

Authors comment: More results have been added in the revised manuscript (Lines 32-39)

Line 37-40 very long sentence consider a re-write

Authors comment: Change has been made in the revised manuscript (Lines 43-45)

Line 42 delete “however”

Authors comment: Change has been made in the revised manuscript (line 51)

Line 43-48 why was not compared to “grass-grass early weaning”?

Authors comment: This has been clarified in the introduction (Lines 45-47)

Line 48 what is the primary driver then?

Authors comment: We actually do not know that and that is why the study was designed, to look at the impacts of prior exposure to Herb-clover mixes to try to get a consistent response. Based on previous studies, reason for this variation is still unknown. That is why lines 64 to 67 outline some potential options

Table 1 I found this table confusing! Perhaps the symbol for “Lambs on herb-clover mix” is missing in the pdf review version? Check this please

Authors comment: This has been corrected in the revised manuscript (Line 121-123)

Line 119 replace “onset of the main experiment” with “early weaning” or similar description

Authors comment: This has been corrected in the revised manuscript (Line 142)

Line 153 delete repeated commas

Authors comment: This has been corrected in the revised manuscript (Line 177)

Line 182 were these samples cut above ground level or expected post-grazing mass?

Authors comment: Samples were cut to ground level but grab samples were taken from within the post grazing mass level. These procedures have been used in previous published studies (see line 207)

Line 242 oh! This is not true! Table 2 shows the opposite

Authors comment: This was an error, corrected in the revised manuscript (Line 267).

Table 2 I understand that some of the weaned ewes joined the commercial flock. So, are these data in table 2 showing what the lambs grazed? What ewes grazed? Or an average of what offered for both

Authors comment: Herbage mass presented in the table was on average offered for both ewes and lambs, additional information has been added to the title of   Table 2, Line 281).

Line 264 delete “in”

Authors comment: This was corrected in the revised manuscript (290)

Line 296 delete “which”

Authors comment: This was corrected in the revised manuscript (line 322).

Line 379-382 any explanation for this? Intake is higher? ME content was even higher for grass than the herb in 2017!

Authors comment: An explanation has been added in the revised manuscript (lines 411-416).

Submission Date

10 June 2020

Date of this review

14 Jul 2020 04:17:25

Round 2

Reviewer 1 Report

The authors have significantly improved the manuscript.

Reviewer 3 Report

Reviewer's comments are satisfactorily addressed